# The Emerging Role of the Aging Process and Exercise Training on the Crosstalk between Gut Microbiota and Telomere Length

**DOI:** 10.3390/ijerph19137810

**Published:** 2022-06-25

**Authors:** Victória Assis, Ivo Vieira de Sousa Neto, Filipe M. Ribeiro, Rita de Cassia Marqueti, Octávio Luiz Franco, Samuel da Silva Aguiar, Bernardo Petriz

**Affiliations:** 1Laboratory of Molecular Analysis, Graduate Program of Sciences and Technology of Health, University of Brasilia, Brasília 72220-275, Brazil; vicassis.assis@gmail.com (V.A.); ivoneto04@hotmail.com (I.V.d.S.N.); marqueti@gmail.com (R.d.C.M.); 2Postgraduate Program in Physical Education–Catholic University of Brasília, Brasília 71966-700, Brazil; filipemouraudf@gmail.com; 3Postgraduate Program in Genomic Sciences and Biotechnology, Proteomic and Biochemical Analysis Center, Catholic University of Brasília, Brasília 71966-700, Brazil; ocfranco@gmail.com (O.L.F.); bernardopetriz@gmail.com (B.P.); 4Laboratory of Molecular Exercise Physiology–Physical Education Department, University Center–UDF, Brasília 70297-400, Brazil; 5Postgraduate Program in Biotechnology, S-Inova Biotech, Catholic University Dom Bosco, Campo Grande 79117-900, Brazil; 6Postgraduate Program in Physical Education–Federal University of Mato Grosso–UFMT, Cuiabá 78060-900, Brazil; 7Postgraduate Program in Rehabilitation Sciences–University of Brasília, Brasília 72220-275, Brazil

**Keywords:** gut microbiota, telomere length, exercise, aging

## Abstract

Aging is a natural process of organism deterioration, which possibly impairs multiple physiological functions. These harmful effects are linked to an accumulation of somatic mutations, oxidative stress, low-grade inflammation, protein damage, and mitochondrial dysfunction. It is known that these factors are capable of inducing telomere shortening, as well as intestinal dysbiosis. Otherwise, among the biological mechanisms triggered by physical exercise, the attenuation of pro-inflammatory mediators accompanied by redox state improvement can be the main mediators for microbiota homeostasis and telomere wear prevention. Thus, this review highlights how oxidative stress, inflammation, telomere attrition, and gut microbiota (GM) dysbiosis are interconnected. Above all, we provide a logical foundation for unraveling the role of physical exercise in this process. Based on the studies summarized in this article, exercise training can increase the biodiversity of beneficial microbial species, decrease low-grade inflammation and improve oxidative metabolism, these factors together possibly reduce telomeric shortening.

## 1. Introduction

Aging is characterized by reduced time-dependent functionality that affects most living organisms, often negatively manifested by chronic diseases and fragility [1]. The global population of older adults over 65 years old in 2019 was approximately 9.09%, and by 2050 it is expected to reach about 20% [2]. Given this, millions of people will age, depending on cultural and socioeconomic conditions [3]. Among factors related to aging are mitochondrial dysfunction, immunosenescence, gut microbiota (GM) dysbiosis, and telomere attrition, resulting in functional disability, multi-morbidities, and poor quality of life [4,5,6,7,8].

Recently, Velando et al. [9] found that gull hatchlings with a microbiome by potential commensal bacteria (e.g., *Catellicoccus* and *Cetobacterium*) display larger telomeres, indicating that a healthy microbiome can protect telomeres and genomic integrity against cellular stress. Furthermore, several pieces of evidence support that the exacerbated reactive oxygen species (ROS) and pro-inflammatory cytokines production lead to greater cell and/or DNA damage, increasing cell division rate and, thus, inducing a greater telomere shortening. In addition, recent research has shown that many GM species could be related to increased oxidative stress and chronic inflammation [8,10,11,12]. Therefore, it is plausible to infer that telomere shortening may be associated with GM dysbiosis through redox imbalance and exacerbated inflammatory signaling pathways. Therefore, strategies for delaying these harmful factors and their main outcomes become necessary.

Otherwise, physical exercise has impacted the gut by increasing microbiome diversity and primary metabolism [13,14]. This means that altering the bacterial composition and its by-products through physical training can be a useful strategy in combating increased oxidative stress and pro-inflammatory cytokines, reducing telomeric attrition. Thus, the relationship between exercise training, gut microbiota, and telomere length during the aging process can gain much attention in the next decade due to its potential use and promise as a future therapeutic target in longevity, disease management, and measurement of genomic aging [15,16].

Here we first discuss the hypothesis of how oxidative stress, inflammation, telomere attrition, and GM dysbiosis are interconnected. In addition, we provide a logical foundation for unraveling the role of physical exercise in this process. Deciphering the crucial role of exercise training in altering gut microbiota and telomere biology in inactivity and age-related progression is a promising issue for continued understanding of how a healthy lifestyle is essential for general health and lifespan, as well as for the prevention and treatment of several chronic diseases inherent to the aging process.

## 2. The Role of the Aging Process and Exercise Training on Intestinal Microbiota during Biological Aging

In healthy humans, the role and composition of GM are physiologically modulated by several environmental factors related to lifestyle, such as physical exercise, position, geography, and eating habits [9,10]. The gut ecosystem of healthy adults demonstrates strong adaptations to potential stressors, such as pharmacological treatments, acute illness, and lifestyle changes (acute or chronic) [11]. However, each type of stressor is capable of generating a signature in the microbial composition regarding the balance between symbiotic and pathobiont microorganisms [12]. Cross-sectional studies of fecal samples from subjects in different age groups indicate age-related changes in GM competition and diversity, supporting longitudinal studies [13]. In general, GM in the elderly is more reduced in terms of biodiversity throughout life. For example, there is an increase in opportunistic Gram-negative bacteria. On the other hand, there is an attenuation of species such as *Bifidobacterium*, *Lactobacilli* and fatty acid producers of short-chain (AGCC), which are related to the maintenance of the health of the host [14,15].

Some variations in GM are much more associated with biological or functional age than chronological age [13,16]. Furthermore, GM biodiversity is inversely correlated with biological age. However, it does not correlate with chronological age [17]. The possible explanation for this phenomenon is that, as biological age increases, the overall richness of GM attenuates, facilitating the proliferation of microbial taxa related to unhealthy aging [17]. For example, centenarians are generally regarded as a biological model of thriving aging due to their ability to live in a somewhat good state of health for several decades. [18,19]. For several studies have pointed out that the composition of the GM of centenarians of various ethnicities and geographical positions presents a high biodiversity and representation of *Firmicutes*, *Bifidobacterium*, as well as SCFA producers, with their respective anti-inflammatory and homeostatic properties, when compared with the elderly or younger adults [20,21,22], supports the concept that gut microbiota behavior does not reflect chronological but biological aging [18].

From another perspective, the GM biodiversity of frail elderly people or those with certain mobility limitations presents certain degrees of dysbiosis, followed by a reduction in species richness and possible imbalance between pathogens and microorganisms with anti-inflammatory properties [17,23]. The most severe cases of dysbiosis were identified in elderly patients with several comorbidities admitted to the hospital, where the action of acute and chronic diseases, drugs, and inactivity become potent disruptors of the balance of intestinal microbial ecology [24,25]. Surprisingly, similar changes were observed in centenarians with deteriorating health status [26]. However, these studies only support the thesis that the composition of GM may reflect each individual’s aging process. However, it has not yet been described to what extent the microbiota is a biomarker of aging [12]. Therefore, the fine line between host health and environmental influences still needs to be understood due to the fact that when considering that GM contributes to the triggering of geriatric giants, such as frailty, sarcopenia, and cognitive impairment [17,27,28]. It is also modulated by the lifestyle, which would, in fact contribute to a healthy or unhealthy aging of the individual.

Although the microbiota is relatively stable throughout adulthood, aging induces significant shifts in gut microbiome composition and function associated with a decline in diversity when compared to young individuals [17]. Such declines are correlated with impaired intestinal barrier function, including declined functions of Paneth cells and IgA-mediated mucosal immunity, while there is an increase in the magnitude of pro-inflammatory cytokines and dysfunctional epithelial tight-junctions permeability [17]. The known microbiome pattern of healthy aging is characterized by a depletion of core genera Bacteroides [18]. Retaining a high Bacteroides dominance into older age can predict decreased survival in a four-year [18]. Ghosh et al. [19] reported that the typical aging-related alterations are typified by a loss of dominant commensal taxa (such as *Prevotella*, *Faecalibacterium*, *Eubacterium rectale*, *Lachnospira*, *Coprococcus*, and the health-associated genus Bifidobacterium). Furthermore, these taxa seemed to be replaced by a second group of commensals (such as the putatively beneficial *Akkermansia*, *Christensenellaceae*, *Butyricimonas*, *Odoribacter* and *Butyricicoccus*) and pathobionts (*Eggerthella*, *Bilophila*, *Fusobacteria*, *Streptococcus*, and *Enterobacteriaceae*). These microbiome modifications are related to health status.

On the other hand, Zhu et al. [20] reported that regular exercise and higher frequency induce an up-regulation in the relative abundance of bacterial functional pathways linked to nucleotide metabolism, glucose metabolism, and lipid metabolism in the active elderly when compared to sedentary individuals. Moreover, the authors demonstrated that microbial functions were partly reestablished by regular exercise in the elderly obese via adjustments in bacterial functional pathways associated with vitamin, nucleotide, and glucose metabolism. Regarding specific bacteria composition, Erlandson et al. [21] found that after a 24-week of combined exercise intervention with a gradual increase in intensity in men older sedentary, there was an increase in the genera Bifidobacterium, *Oscillopsira*, and *Anareostipes*, while decrease *Oribacterium* and *Prevotella*. In a recent systematic review, it was observed that different exercise modalities (aerobic and resistance training) may act through changes in the gut environment such as rises in the production of short-chain fatty acids, bile acids alteration, lipopolysaccharide reduction, besides regulation of mucus production which may, in turn, change the composition of the gut microbiome towards a healthier one by promoting the growth of health-related bacteria, and a decrease in harmful bacteria [22].

In an aging animal model, Piao et al. [23] observed that the pathologic changes of intestinal senescence (gut inflammation, extensive changes in fecal metabolomic profiles, and marked gut flora disorders) were associated with the decrease of telomerase mRNA in elderly mice, which suggests a telomere-gut connection. Furthermore, there is evidence that the gut is one of the earliest organs to exhibit short telomeres and tissue dysfunction during normal zebrafish aging [24]. Current, Velando et al. [9] discovered that gull hatchlings with a microbiome by potential commensal bacteria (e.g., *Catellicoccus* and and *Cetobacterium*) exhibit larger telomeres, suggesting that a healthy microbiome can protect telomeres and genomic integrity against cellular senescence. From another perspective, previous studies have displayed those certain foods contribute to healthy gut microbiota and consequently influence telomere length [25]. For example, individuals who consume higher amounts of nuts, seeds, fruits, and vegetables have longer telomere lengths than others. In contrast, processed meats were associated with and harmful intestinal bacteria and shorter telomere length [25]. In addition, a recent investigation reported an association of the gut microbiota composition with leukocyte telomere length in young individuals [26].

## 3. Telomeres: Structure and Function

Telomeres, from the Greek telos ‘end’ and mere ‘part’, are DNA-protein structures found at the end of chromosomes that are critical to maintaining the stability and integrity of the genome [27,28,29,30,31]. In human cells, telomeres comprise between 10 and 15 kilobases of a highly conserved hexameric tandem repeat (TTAGGG) DNA sequence [27,28,29,30,31]. Telomeres are marked by the presence of 30 to 400 nucleotide-long 3′ overhang of a G-rich strand. The G-strand overhang can fold back and invade the double-stranded telomeric region, forming a so-called T-loop [28,30]. This is organized and associated with specialized proteins, including, among others, the Shelterin complex, which is composed of six proteins: telomeric repeat binding factors 1 and 2 (TRF1 and TRF2), TRF1-interacting protein 2 (TIN2), protection of telomeres protein 1 (POT1), TIN2, POT1-interacting protein (TPP1), and repressor/activator protein 1 (RAP1), which, together with telomerase activity, have crucial functions in regulating telomere length and protecting telomeres from the DNA damage response [27,28,29,30,31].

Telomeres shorten in each cell division, which is usually due to the end replication problem, i.e., the inability of the DNA replication machinery to complete the synthesis of the tip of the chromosomes. Telomere shortening also occurs through oxidative damage and other end-processing events in dividing and non-dividing cells. Dysfunctional telomeres, caused by telomere shortening, the collapse of telomere structure, or displacement of shelterin complexes from telomeres, trigger a DNA damage response and loss of cell proliferation leading to senescence or apoptosis [27,28,31]. In addition, telomere shortening can be influenced by genetic factors (e.g., TERT and TERC, genes important for telomeric maintenance), epigenetic factors, and other factors such as age, gender, body fat, ethnicity, inflammation, socioeconomic factors, and physical activity level [28,32,33,34].

## 4. The Possible Crosstalk between Gut Microbiota and Telomere Length: Involvement of Oxidative Stress and Inflammation

Initially, progressive chromosome shortening occurs during cell replication and is observed with the aging process. Senescent cells accumulate with age, secrete inflammatory cytokines, and have well-established roles in promoting degenerative diseases and pathology with aging [35]. In addition, many of the environmental exposures (sedentary lifestyle, unhealthy diet, drugs, smoking) that lead to ROS elevation, are also associated with shortened telomeres [35]. Importantly, due to the fact that dysfunctional mitochondria during aging generate more ROS, this sets up a vicious cycle in which telomeres may suffer further insult from oxidative damage. Accumulating evidence from human and animal models suggests that oxidative damage to telomeric DNA is in charge of accelerated telomere shortening under oxidative stress [35]. These injuries include damaged single-strand breaks, purines, pyrimidines, and a basic site [35]. Guanine is the most susceptible of the natural bases to oxidation, commonly generating 8-oxoguanine, which is even more sensitive to oxidation. Furthermore, there is data that shows the antioxidant peroxiredoxin 1, which scavenges H_2_O_2_, is enriched at telomeres, and PRDX1 loss inherent to the aging process starts to telomeres damage [36]. At last, the habitual genomic DNA damage repair mechanisms (base excision, nucleotide excision repair, mismatch repair, homologous recombination and non-homologous end joining) are not efficient during aging.

In addition, several species of GM in pathological contexts are related to increased oxidative stress and inflammation [37]. Increased abundance of several bacteria species such as *Enterotoxigenic B. fragilis, Fusobacterium varium, Adherent-invasive Escherichia coli, Campylobacter concisus,* and *Fusobacterium nucleatum* is linked with inflammatory diseases such as inflammatory bowel disease (IBD) and metabolic-associated fat liver disease (MAFLD), respectively [38,39,40,41]. Oxidative stress-mediated by the human microbiome occurs through its ability to alter the cellular ROS by modulating mitochondrial activity [42]. It has been hypothesized that GM may be related to telomere shortening.

Some bacterial species synthesize enzymes capable of fermenting non-digestible Polysaccharide into digestible compounds, known as short-chain fatty acids (SCFAs) [43] this sense; it’s known that *Bifidobacteria longum* and *Eubacterium rectale* degrade indigestible oligosaccharides into acetate and butyrate, two of the three most common SCFAs [43]. These gut-derived metabolites are linked to the regulation of skeletal muscle function and metabolism [44]. Thus, the process of SCFAs in old age is multifaceted, and its potential role in the prevention of sarcopenia is currently being discussed [45]. Furthermore, SCFAs have been linked to reduced oxidative stress and inflammation. Huang et al. [46] pointed out that SCFAs, mainly acetate, butyrate, and the GPR43 agonist, could attenuate oxidative stress and inflammation of glomerular mesangial cells (GMCs) induced by high glucose and LPS circulation. In addition, they reduced the generation of ROS and pro-inflammatory cytokines (MCP-1, IL-1ÿ, and ICAM-1). Thus, although further studies are needed to define better the mechanisms by which the microbiota can reduce oxidative stress and inflammation, it is already possible to consider the existence of this cross-talk [46].

The intestinal microbiota can also be modulated by antioxidant metabolism; for example, antioxidant substrates from fermented foods can play significantly essential roles in reducing oxidative stress by hindering lipid peroxidation [47]. The *Lachnospiraceae* and *Ruminococcaceae* families can produce reactive sulfur species (RSS) and enhance the antioxidant capacity of the host [48]. Aging is commonly characterized by low antioxidative defense systems, specifically in glutathione peroxidase levels, the most potent biological antioxidative reductant [49]. In agreement, excessive ROS production (H_2_O_2_, singlet oxygen, and others) has been observed in aging [50]. Thus, it seems that the gut microbiota may also influence the telomere shortening process by participating in the balance of the antioxidant profile and ROS levels in cells.

## 5. The Emerging Role of Exercise on Gut Microbiota and Telomere Length during Biological Aging

Regular physical exercise is a well-established strategy to combat and prevent morbidity and mobility risks [51]. Physical activity stimulates several metabolites and inflammatory mediators’ production [52,53]. Nevertheless, a regular and well-structured exercise program (e.g., moderate to vigorous intensities) with adequate rest can suppress the circulation of basal pro-inflammatory cytokines, which may indicate a regulatory loop between exercise-induced physiology and one’s immunity [52]. Therefore, exercise promotes the circulation of several anti-inflammatory signals capable of reducing systemic inflammation, reducing macrophage infiltration in adipose tissue [54], attenuating the expression of Toll-like receptors (TLR2 and TLR4) in immune cells [55], decreasing M1 and CD8+ macrophages [56], among other signaling pathways to reduce inflammation [57,58,59].

From another perspective, moderate-intensity exercise mitigates the impacts of stress-triggered intestinal barrier dysfunction. It is related to intestinal permeability reduction in lower bacterial translocation rates and conservation of mucosal thickness [53]. One possible hypothesis of how exercise can influence the microbiota is by increasing core temperature and causing heat stress, mainly when performed for long periods [60]. Exercise also reduces intestinal blood flow by more than 50%, and significant intestinal ischemia can occur within ten minutes of high-intensity exercise [61].

Although the gut is an anaerobic environment, intestinal epithelial cells primarily utilize oxidative metabolism, and high-intensity exercise is known to temporarily impair intestinal barrier function [61,62]. Therefore, exercise-induced heat stress and ischemia may, in the short term, lead to more direct contact of the gut mucosal immune system with gut microbes, which may impact the gut microbial community. Furthermore, such changes in GM promoted by physical exercise seem to be more significant in early life when compared to adulthood [63]. An interesting study performed a 12-week intervention in healthy older women to address the effects of different exercise methods. At the end of the study, women assigned to brisk walking as aerobic training showed increased *Bacteroidetes* and a decrease in *the Clostridium XIV subgroup.* In contrast, a control group that did only trunk muscle training showed an increase in *Clostridium group IX*. Aerobic training seemed to be able to promote significant changes in bacterial composition, mainly in increasing the abundance of *Bacteroidetes*, which may be correlated with an improvement in the cardiorespiratory fitness of elderly women [64].

Regarding preserving telomere length, the main effects triggered by exercise are reducing oxidative stress [65] and the inflammatory profile [66]. Exercise modulates the mutual crosstalk between oxidative stress and inflammatory state through factor nuclear kappa B (NF-kB) signalizing pathways suppression [67]. Such decreases were observed as H_2_O_2_ modulates IKK-dependent NF-kB activation by promoting the redox-sensitive activation of the PI3K/PTEN/Akt and NIK/IKK pathways [67]. There is a significant decrease in the chemotaxis of immune cells after chronic exercise, which guides antioxidant agents’ bioavailability [68]. Moreover, exercise training decreases pro-inflammatory cytokine levels and reduces ROS production, mitigates DNA damage, genomic instability, and the apoptosis process. Even though studies indicate that increased oxidative stress causes impairment of DNA repair mechanisms and telomere wear [69,70]. The relationship between telomere length and oxidative stress appears to be well balanced, where oxidative stress can help maintain telomeres. Still, this level needs to be below certain thresholds to reduce damage. In addition, inflammation can also shorten telomeres; as leukocyte volume increases, hematopoietic stem cell division is activated, which increases cell replication and subsequently leads to telomere shortening [71]. The pro-inflammatory cytokine tumor necrosis factor (TNF)-α can also shorten telomere by down-regulating telomerase [72]. The shortening of leukocyte telomeres may be associated with increased levels of interleukin (IL)-6 and TNF-α. Furthermore, subjects with high IL-6 and TNF-α were likelier to have shortened leukocyte telomeres than subjects with higher levels of just one of these molecules [73]. On the other hand, this length can vary according to physical activity levels and age. In an elegant study, Werner et al. [74] compared the telomeres of young and middle-aged endurance athletes with sedentary controls. Authors found that regular training reduces leukocyte telomere shortening [74]. Furthermore, the study showed that the telomeres of middle-aged athletes were preserved, as were young controls. In contrast, the leukocyte telomere length of middle-aged controls was greater than that of younger controls, suggesting that the reduction is correlated with age [74] and exercise intensity [75]. Corroborating this perspective, another recent study by Aguiar et al. showed that master athletes reduced the aging profile related to preserved DNA sequencing compared to age-matched control. This may be associated with lower oxidative stress, lower chronic inflammation, and higher telomerase activity [76].

Regarding possible mechanisms, upregulation of shelterin proteins and telomerase activity may explain how exercise protects against telomere shortening. It was demonstrated that young and aged athletes had higher expression of TRF2 mRNA, greater telomerase activity, and lower expression of Chk2 mRNA than young controls [74], which indicates that exercise has positive effects on telomere structure cellular fate. Similarly, previous studies demonstrate that exercise training can increase shelterin expression in the heart, skeletal muscle, aorta, and large intestine of aged rodents [77,78], reinforcing our hypothesis that gut microbiota potentially influences telomere length. Schematic illustration of how the gut microbiota influence telomere length during aging via exercise is cited in Figure 1.

Schematic illustration of a hypothetical mechanism by which physical exercise modulates crosstalk between gut microbiota and telomere length is reported in Figure 1. Possibly, the physical training can increase the beneficial microbial species proportion, microbial biodiversity, greater production of improved short-chain fatty acids, and carbohydrate synthesis and metabolism, which consequently mitigate local mitochondrial dysfunction, apoptosis, and cell death.

Consequently, these adaptations inherent to the exercise training can mitigate the oxidative, stress, mitogenic signals, low-grade inflammation, and impaired resistance to molecular signals stressors. In contrast, increasing possible can mitigate the oxidative, stress, mitogenic signals, low-grade inflammation, and impaired resistance to molecular signals stressors, while increasing protein shelter and telomerase activity. Everything would indicate that these factors would make telomeres less shortened, generating minor DNA damage, less genomic instability, and overall protein homeostasis.

## 6. Current Perspectives

Considering the above, future research should be focused on developing experimental studies to confirm the relationship between exercise training, gut microbiota, and telomere length during the aging process. Agreed the complexity of biological systems during aging, the employ of experimental animal models might provide a meaningful understanding of the several adaptive mechanisms inherent to acute and chronic exercise that potentially preserves gut microbiota and telomere integrity, mainly when ethical considerations limit the application of human studies. Morphological, biochemical, and molecular approaches might aid knowledge of the detailed picture of cellular adaptation in response to the exercise, besides the further discovery of potential therapeutic targets. A mechanistic framework of such responses could give valuable insights into therapeutic approaches for development and treatment guidance in the elderly and clarify the main constraints and positive effects of exercise in different physiological systems.

We suggest randomized controlled trials comparing different exercise modalities (e.g., aerobic, resistance exercise, and combined) on molecular pathways involved in gut microbiota, telomere length, and other age- or disease-related outcomes. Furthermore, those involved with exercise prescription, including coaches, rehabilitation specialists, and exercise physiologists, must understand the acute program variables to potentiate training adaptations. Hence, investigations involving distinct exercise frequency, volume, and intensity at various time points could elucidate the potential mechanisms in the relationship between gut microbiota and telomere biology. Finally, whether these effects are tissue-specific or systemic remains a provocative hypothesis for further investigation. For innovative molecular advances, a blend of metabolomics with transcriptomics and proteomics approaches in exercise-related effects can be essential for discovering a novel regulatory mechanism that mediates the communication between gut microbiota and telomere length. Furthermore, research in fecal microbiota transplantation may be an exciting approach to elucidate the possible relationship between intestinal microbiota and telomeric preservation.

The emerging role of exercise training on the crosstalk between gut microbiota and telomere length adds to a debate on drug development to potentiate the chronic effects of exercise. Indeed, the poor prognosis during aging highlights the need for developing novel pathways for testing new drugs and therapeutic approaches. Based on the studies summarized in this paper, the drug’s progress must necessarily focus on novel approaches to inhibition of inflammatory signaling pathways factors (Toll-like receptors, proinflammatory cytokines). Moreover, a possible target for new drugs may be activating several antioxidants signaling pathways (Nuclear Factor Erythroid 2-Related Factor 2, glutathione peroxidase, SOD, and catalase) to minimize gut microbiota dysbiosis and telomere dysfunction. Finally, the interference of gut microbiota on telomere length can lead to health or human performance solutions, helping design new supplements and probiotics that combat age-related changes.

## 7. Conclusions

It is quite suggestive of supposing that the mechanisms of aging and longevity are distinct paths that modulate the lifespan of all beings. However, biological aging is a multi-factorial and complex process where the physiological systems intertwine. Thus, more detailed research into how the crosstalk between the microbiome and telomere integrity influences cellular senescence and understanding which exercise characteristics enhance this process can be essential for overall health and longevity. Defining the mechanisms that mediate this interaction is essential in developing optimized lifestyle interventions to reduce chronic disease risk during aging. We hope our insights stimulate future research and more open debate about the relevance of exercise on the gut microbiota and telomere in the aging context.

## Figures and Tables

**Figure 1 ijerph-19-07810-f001:**
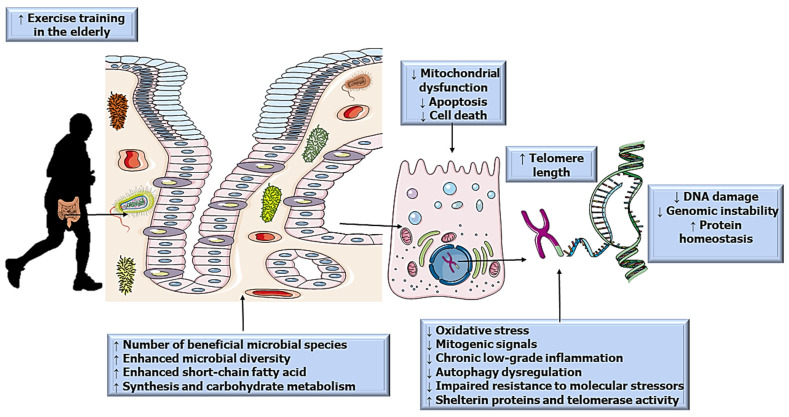
Schematic representation of the hypothetic mechanisms of how exercise training modulates crosstalk between gut microbiota and telomere length.

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
