# Peer review of "The Emerging Role of the Aging Process and Exercise Training on the Crosstalk between Gut Microbiota and Telomere Length"

_ijerph, 2022, doi:10.3390/ijerph19137810_

Round 1

Reviewer 1 Report

Review for the manuscript Does the gut microbiota influence telomere length? The emerging role of exercise on biological aging

            Dear Editor, thank you very much for the invitation to review this manuscript wich is very interesting.  Please, find below my suggestions to the authors.

First of all I suggest changing the title “Does the gut microbiota influence telomere length? The emerging role of exercise on biological aging”. What is the article about? Role of microbiota on telomere length or the role of the exercise on biological aging? Or both? I understand the relationship that is also explained later in the text. However, the title, in the form it is, is confusing for the readers.

ABSTRACT

I suggest that Abstract brings more results than background as we see in the current form.

INTRODUCTION

In lines 45 and 46 we can find “Recently, Velando et al. [5] found that gull hatchlings with a microbiome by potential commensal bacteria (e.g., Catellicoccus and Cetobacterium) display larger telomeres…”. Please, do not use italics for “and”.

In lines 57-62 we can see “This means that altering the bacterial composition and its by-products through physical training can be a useful strategy in combating increased 58 oxidative stress and pro-inflammatory cytokines, reducing telomeric attrition. Thus, the relationship between exercise training, gut microbiota, and telomere length during the  aging process can gain much attention in the next decade due to its potential use and promise as a future therapeutic target in longevity, disease management, and measurement of genomic aging”.

Please include references in this paragraph.

Gut microbiome, exercises, telomeres and aging are very relevant and current topics. I miss the Introduction, new articles that talk about these subjects. For example, see studies 10.1038/s41598-022-12578-7 / 10.1016/j.neures.2022.05.003 / 10.3390/nu14091881. / 10.3390/nu14091758 and many others. Therefore, I also believe that a review that considers these matters should contain many more references than just 42.

In line 81-83- we can find “Increased abundance of several bacteria species such  as Enterotoxigenic B. fragilis, Fusobacterium varium, Adherent-invasive Escherichia coli, Cam-pylobacter concisus, and Fusobacterium nucleatum is linked with inflammatory diseases such  as inflammatory bowel disease (IBD), ulcerative colitis (UC)…”

 Ulcerative colitis is an Inflammatory Bowel Disease. Do not use it separated here because it seems it is a different entity.

Please, see in line 84: “…Non-alcoholic steatohepatitis (NASH)…”
           Non-alcoholic steatohepatitis (NASH) is now suggested to be named MAFLD (Metabolic-Associated Fat Liver Disease). Please include the definition proposed by
Nahum Méndez-Sánchez et al. 2022 (doi: 10.1016/S2468-1253(22)00062-0).

In line 94: The authors say that “Furthermore, SCFAs inhibit oxidative stress and inflammation…”. I suggest to explain how SCFAs inhibit OS and inflammation.

In the section 3:

Here we can read (lines 109-116) “Regular physical exercise is a well-established strategy to combat and prevent morbidity and mobility risks [23]. Physical activity stimulates several metabolites and inflammatory mediators production [24,25]. Nevertheless, a regular and well-structured exercise program (e.g., moderate to vigorous intensities) with adequate rest can suppress the circulation of basal pro-inflammatory cytokines, which may indicate a regulatory loop between exercise-induced physiology and one's immunity [24]. For example, moderate-intensity exercise mitigates the impacts of stress-triggered intestinal barrier dysfunction. It is related to intestinal permeability reduction in lower bacterial translocation rates and conservation of mucosal thickness [25]”. I think it would be worthy to include which are the inflammatory mediators mentioned above. Inflammatory process has a key-role in the aging process. For these reasons, not mentioning the pro-inflammatory biomarkers will lead this sentence to be vague.

I have the same impression after reading the sentence (lines 135-137) “Regarding preserving telomere length, the main effects triggered by exercise are reducing oxidative stress [31] and the inflammatory profile [32].” What is the inflammatory profile? How does the process interfere with oxidative stress?

In lines 170-171: This sentence “Schematic illustration of a hypothetical mechanism by which physical exercise modulates crosstalk between gut microbiota and telomere length.” Refers to Figure 1? Please review. It seems to be “alone” in this paragraph.

Please, also review the  sentence (lines 171-172)“For there is evidence that indicates a correlation between physical training and microbiota and physical training  and telomeres”. For there is?

I also suggest reviewing and re-writing these sentences (lines 173-176): “Thus, the present hypothesis has this possible communication between the 173 variables as its central point since physical training can increase the proportion of beneficial microbial species, microbial biodiversity, greater production of improved short-chain fatty acids, and carbohydrate synthesis and metabolism. What at cellular levels should present less mitochondrial dysfunction, apoptosis, and cell death..”

In lines 178-182 we can read “Thus, when considering the possible ability not only of exercise, but also of the microbiota to modulate oxidative metabolism and reduce this stress, mitogenic signals, low grade inflammation, impaired resistance to molecular stressors, and increased protein shelter and telomerase activity”. I think there is something missing in this sentence. Please re-write.

CONCLUSION
I suggest removing “valuable” form the conclusion of the authors.

In the Introduction section (lines 66-68) the authors say that “Here, we present vital mechanisms, major research tracks, and critical gaps in this emerging area, providing essential discussion and valuable conclusions about healthy aging.” I must disagree with this statement because in the Conclusions section, the authors posed more challenges for further work and research than they actually drew valuable conclusions. 

Reviewer 2 Report

The author's review is an interesting topic and can be evaluated in that it provides insights into effects of exercise and gut bacteria on telomeres, in the view of aging factors such as oxidative stress, inflammation, and telomere. The manuscript is well summarized, however there are several concerns against the discussion. The comments on the manuscript are as follows;  

1. For the general reader, please specify the effect of oxidative stress on DNA and telomere length. Please explain biologically.

2. Please discuss whether the decrease in telomere length due to aging is due to the number of divisions or aging damage.

3. Please add a description of how the intestinal flora changes with age.

4. Similarly, how does exercise change the gut microbiota?

5. How does exercising at aging have a different effect on the gut microbiota than at a young age?

6. There is little debate about the specific effects of gut bacteria on telomeres. Please add a discussion about the specific effect on the "length" of telomeres, citing previous studies.

7. Overall, the names of gut microbiota are generally not familiar. I expect a brief introduction as well as a scientific name.

8. Line237, what is “o”? how the “o” crosstalk… maybe typo?

Round 2

Reviewer 1 Report

Dear authors,
The manuscript is now ready for publication.